# Bis-Oxadiazole Assemblies as NO-Releasing Anticancer Agents

**DOI:** 10.3390/pharmaceutics17111494

**Published:** 2025-11-19

**Authors:** Egor M. Matnurov, Irina A. Stebletsova, Alexander A. Larin, Jemma Arakelyan, Ivan V. Ananyev, Artem L. Gushchin, Leonid L. Fershtat, Maria V. Babak

**Affiliations:** 1Drug Discovery Lab, Department of Chemistry, City University of Hong Kong, 83 Tat Chee Avenue, Hong Kong SAR, China; ematnurov2-c@my.cityu.edu.hk (E.M.M.); jemmaarakelyan93@gmail.com (J.A.); 2N.D. Zelinsky Institute of Organic Chemistry, Russian Academy of Sciences, 47 Leninsky Prosp., 119991 Moscow, Russia; irinastebl@icloud.com (I.A.S.); roby3@mail.ru (A.A.L.); 3Higher Chemical College of the Russian Academy of Sciences, D.I. Mendeleev University of Chemical Technology of Russia, 9 Miusskaya Square, 125047 Moscow, Russia; 4N.S. Kurnakov Institute of General and Inorganic Chemistry, Russian Academy of Sciences, GSP-1, Leninsky prospect, 31, 119991 Moscow, Russia; ananyev@ineos.ac.ru; 5A.V. Nikolaev Institute of Inorganic Chemistry, Siberian Branch of Russian Academy of Sciences, 3 Academician Lavrentiev Ave., 630090 Novosibirsk, Russia; gushchin@niic.nsc.ru

**Keywords:** nitrogen heterocycles, furoxan, oxadiazole, malignant pleural mesothelioma, NO donors, antiproliferative activity, mechanism of action

## Abstract

**Background**: Malignant pleural mesothelioma (MPM) is an aggressive, asbestos-associated cancer characterized by dysregulated nitric oxide (NO) signaling and increased NO levels that facilitate tumor progression. Paradoxically, this aberrant NO environment creates a therapeutic vulnerability that can be exploited by NO-donor prodrugs, which overwhelm cellular defenses with cytotoxic concentrations of NO, inducing nitrosative stress and apoptosis. Within this framework, oxadiazole-based scaffolds have emerged as a promising platform for prodrug development owing to their versatile chemistry and potential as novel NO donors or synergistic agents. In our previous studies, we developed several series of hybrid architectures incorporating 1,2,5-oxadiazole 2-oxide (furoxan) and 1,2,4-oxadiazole scaffolds, producing compounds with diverse and tunable NO-donor activities. We further observed that the cytotoxicity of these hybrids was significantly influenced by the substituents introduced at position 3 of the furoxan ring. **Methods**: We designed and synthesized a series of bis(1,2,4-oxadiazolyl)furoxans to systematically investigate their NO-donating capacity, cytotoxicity against MPM cell lines, selectivity over healthy lung fibroblasts, and underlying anticancer mechanisms. **Results**: The bis(1,2,4-oxadiazolyl)furoxans exhibited lower overall cytotoxicity but significantly higher selectivity compared with previously studied 3-cyano-4-(1,2,4-oxadiazolyl)furoxans. Their NO-releasing properties showed a strong correlation with their ability to induce mitochondrial damage, as evidenced by membrane depolarization. Moreover, the incorporation of specific substituents, such as a furan ring, on the 1,2,4-oxadiazole moiety introduced an additional mechanism of action through the induction of reactive oxygen species. **Conclusions**: Analysis of cancer cell death confirmed that these compounds acted through a multimodal mechanism dependent on both NO release and the specific substituents on the 1,2,4-oxadiazole moiety.

## 1. Introduction

Cancer remains the second leading cause of death worldwide, following cardiovascular diseases. Current treatment modalities include surgery, radiation therapy, chemotherapy, gene therapy, and immunotherapy. Despite the high efficacy of these approaches, cancer cells frequently develop drug resistance, highlighting the urgent need for new anticancer drugs and therapeutic candidates [1,2,3].

From a medicinal chemistry perspective, nitrogen-containing heterocycles represent some of the most prevalent scaffolds in modern pharmaceuticals [4,5,6,7]. According to the U.S. FDA database, over 59% of approved small-molecule drugs for clinical use contain at least one nitrogen heterocyclic subunit [8,9]. The widespread occurrence of various nitrogen heterocyclic systems across numerous clinical drugs with different action mechanisms is primarily due to the presence of heteroatoms, which enable more efficient binding to target proteins in living organisms. Therefore, owing to their increased affinity for receptors and bioavailability, pharmacologically oriented molecular systems containing heterocyclic fragments represent one of the most promising classes of organic compounds in the search for new therapeutic agents [10].

Among pharmacologically active compounds, heterocyclic nitric oxide (NO) donors have emerged as an important subclass in organic and medicinal chemistry [11,12,13,14,15]. For decades, classical NO donors such as nitroglycerin and nitroprusside have been used clinically to treat cardiovascular disorders [16]. Beyond these traditional applications, NO donors have also demonstrated potential in managing other pathological conditions, including cancer, by inducing apoptosis in malignant cells [13,15]. Unlike the widely known nitroglycerin, heterocyclic NO donors exhibit hydrolytic stability, do not promote nitrate tolerance, and possess improved pharmacological profiles. A diverse range of heterocyclic NO donors—including 1,2,5-oxadiazole *N*-oxides (furoxans) [17,18,19,20,21,22], 1,2,4-oxadiazoles [23,24], azasydnones [25,26], sydnone imines [27], triazole oxides [28], and pyridazine dioxides [29]—have been synthesized and evaluated for their pharmacological properties (Figure 1A). Notably, furoxan-based derivatives have shown significant promise as anticancer agents for the selective treatment of various types of cancer [30,31,32,33].

Several studies have demonstrated that combining furoxans with other pharmacophoric scaffolds can enhance anticancer activity and increase the sensitivity of drug-resistant cancer cells to treatment [34,35,36]. In our previous work, we investigated the effects of hybridizing furoxans with various heterocyclic systems on their cytotoxicity (Figure 1B). Molecular hybridization of the furoxan ring with nitrogen-containing heterocycles such as pyridine, triazole, or triazine was found to enhance the anticancer activity of the resulting hybrids against multiple human cancer cell lines [37]. Furthermore, 4-amino-3-(indenotriazin-3-yl)furoxan induced apoptosis through caspase-3/7 activation in chronic myeloid leukemia K562 cells, highlighting its strong anticancer potential [38].

In our previous work, we synthesized diverse libraries of furoxan-based hybrid compounds incorporating multiple oxadiazole rings derived from readily available amidoxime precursors **1a**–**1c** and evaluated their cytotoxicity after 72 h of incubation (Figure 2) [37,39]. Notably, nitrobifuroxans **Ia,b** exhibited potent anticancer activity against cervical (HeLa) and rhabdomyosarcoma (RD) cancer cells (IC_50_ = 0.53–21.44 µM), with 3–39-fold selectivity over non-cancerous HEK293 cells [37]. In contrast, replacing the nitrofuroxan moiety with a 1,2,4-oxadiazole ring led to a marked reduction in anticancer activity, as indicated by the in vitro screening of compounds **II** (IC_50_ > 100 µM) [37]. However, replacing the methyl derivatives **II** with cyano derivatives **III** produced a more favorable pharmacological profile, demonstrating strong antiproliferative effects (IC_50_ = 0.9–7.0 µM) against malignant pleural mesothelioma (MPM) AB1 and JU77 cell lines, along with up to 2.5-fold selectivity toward non-cancerous MRC-5 lung fibroblasts [39]. Interestingly, the introduction of a second 1,2,4-oxadiazole fragment produced inconclusive results. Bis(1,2,4-oxadiazolyl)furoxan **2a** exhibited strong anticancer activity (IC_50_ = 2.37–8.75 µM) against A549, HCT116, HeLa, MCF7, and RD cell lines, with at least a 2-fold selectivity toward non-cancerous HEK293 cells (IC_50_ = 20.44 µM). In contrast, its 4-nitrophenyl derivative **2b** showed no cytotoxic effects across the same cell lines [37]. Motivated by these observations, we aimed to elucidate the structure–activity relationships of these triheterocyclic compounds. Accordingly, in this study, we report the synthesis and biological evaluation of a new series of bis(1,2,4-oxadiazolyl)furoxans as promising anticancer agents active against MPM.

## 2. Results and Discussion

### 2.1. Synthesis

For the synthesis of the target bis(1,2,4-oxadiazolyl)furoxans, we employed a synthetic approach based on the reaction of readily available furoxanylbis(amidoxime) **1c** with in situ activated carboxylic acids, except for compound **2a**, which was synthesized using orthoformate in the presence of boron trifluoride etherate. Specifically, activation of the corresponding carboxylic acid with 1,1′-carbonyldiimidazole, followed by the addition of bis(amidoxime) **1c** and 1,4-diazabicyclo [2.2.2]octane (DABCO) as a base to promote the heterocyclization step, afforded the desired heterocyclic assemblies **2b–l** (Figure 1, Appendix A). All synthesized bis(1,2,4-oxadiazolyl)furoxans **2** were obtained in good yields, regardless of the electronic nature of substituents on the starting carboxylic acids, demonstrating that this protocol provides a convenient and straightforward route for constructing the 1,2,4-oxadiazole ring. An exception was observed for the dipyridyl derivative **2f**, which was isolated in a lower yield, likely owing to its increased water solubility. The stability of the synthesized heterocyclic compounds was further examined in DMSO-d_6_ via ^1^H NMR spectroscopy. The representative compound **2j** was completely stable in solution for at least 120 h. The corresponding set of ^1^H NMR spectra recorded for **2j** after 1, 20, 48, and 120 h is provided in the (Appendix A).

### 2.2. X-Ray Crystallography and Density-Functional Theory Calculations

The structure of compound **2f** was confirmed by X-ray diffraction analysis (Figure 3, Appendix A). Both symmetry-independent molecules of **2f** exhibit nearly planar conformations of the 1,2,4-oxadiazolyl–pyridine fragment, as expected, with the rotation of the pyridine ring relative to the 1,2,4-oxadiazole moiety not exceeding 6°. In contrast, the rotation of the furoxan ring relative to the 1,2,4-oxadiazole plane is more pronounced (10.9(2)–22.1(2)°), slightly deviating from the trend typically observed for substituted (1,2,4-oxadiazolyl)furoxans. A search of the Cambridge Structural Database reveals that the linked furoxan and 1,2,4-oxadiazole cycles are generally co-planar, with an average interplanar rotation of approximately 10.9°. However, density-functional theory calculations of the equilibrium geometry of **2f** in the gas phase indicate the role of media effects in the flattening of the **2f** molecule in the crystal state. In the gas phase, the rotation of the furoxan ring relative to the 1,2,4-oxadiazole plane reaches 41.3°, resulting in relatively large root-mean-square deviations for the best overlap between the crystal and simulated gas-phase conformations (0.535 and 0.597 Å for the two independent molecules; see Appendix A). Notably, at the PBE0-D3/def2TZVP level of theory, the electronic energy difference between the crystal and gas-phase conformations is 4.5 and 5.1 kcal/mol for the two molecules of **2f**.

Analysis of the crystal packing of compound **2f** reveals multiple strong intermolecular interactions that likely stabilize its otherwise unfavorable molecular conformation. The crystal structure of **2f** exhibits a complex layered arrangement (Figure 4), characterized by the alternation of similar layers, each composed of independent molecules of the same type. Intralayer aggregation of **2f** molecules (Figure 5A) is stabilized by several key interactions: (1) π…π stacking between pyridine and 1,2,4-oxadiazolyl cycles (the shortest interatomic contact is 3.185 Å); (2) O…π interaction between the exocyclic oxygen atom of the furoxan ring and the 1,2,4-oxadiazolyl fragment (the distance between the O1 atom and the corresponding centroid is 2.821 and 2.921 Å for the two independent molecules); (3) hydrogen bonding between the pyridine ring fragments and nitrogen or oxygen atoms of the 1,2,4-oxadiazolyl moiety (C…N distances of 3.506 and 3.515 Å, and C…O distances ranging from 3.302 to 3.569 Å). Further aggregation of layers composed of independent molecules of different types is mediated by CH…N hydrogen bonds formed between pyridine rings (Figure 5B), with corresponding C…N distances ranging from 3.313 to 3.567 Å.

### 2.3. NO Release

Next, we investigated the time-dependent NO release profiles of all synthesized bis(1,2,4-oxadiazolyl)furoxans **2a**–**l** and precursor **1c** using the Griess assay, a well-established and reliable method for the quantitative measurement of NO-donor capacity [38,39,40,41,42]. The release of NO was monitored through its oxidation to the nitrite anion (NO_2_^−^), whose concentration was determined colorimetrically under physiological conditions. The ratio of the measured NO_2_^−^ concentration to the initial concentration of each furoxan derivative **2b**–**l** (100 µM) was used to evaluate their NO-donating ability (Table 1). For comparison, previously reported time-dependent data for compound **1b** and two well-known furoxan derivatives, namely 3-carbamoyl-4-(hydroxymethyl)furoxan (CAS-1609) and 4-ethoxy-3-phenylsulfonylfuroxan (CHF-2363), were included, as these are among the most studied furoxan-based agents currently in clinical trials [39,43].

The ability of bis(1,2,4-oxadiazolyl)furoxans to release NO within 1 h of incubation (35–75%) was 2–3 times those of reference furoxan-based drug candidates CAS-1609 (27%) and CHF-2363 (26%). The amount of NO released remained nearly constant over time for most of the examined bis(1,2,4-oxadiazolyl)furoxans, as shown by the similar values obtained after 1, 24, and 48 h of incubation. This observation differs from the recently reported behavior of 3-cyano-4-(1,2,4-oxadiazolyl)furoxans (III) [39], which are known to be potent NO donors owing to the strong electron-withdrawing effect of the cyano group. Similar to class III, compounds 2a, 2f, and 2g demonstrated increased NO donor capacity with longer incubation periods. As the total amount of released NO for these three compounds exceeded 100%, it can be inferred that the 1,2,4-oxadiazole ring also contributes to NO donation, consistent with previous findings [44,45]. Overall, the most effective NO donors among all tested compounds were the 4-pyridyl- (**2f**)*,* methoxymethyl- (**2g**), and 4-methoxyphenyl- (**2i**) substituted bis(1,2,4-oxadiazolyl)furoxans, whereas other derivatives were less active. In contrast, 3-cyano-4-(1,2,4-oxadiazolyl)furoxans (**III**) exhibit enhanced NO-donor activity in derivatives bearing nitrophenyl or furyl substituents on the 1,2,4-oxadiazole ring.

Based on the obtained experimental data and previous studies [37,39], we propose a thiol-dependent mechanism for NO release from bis(1,2,4-oxadiazolyl)furoxans **2a–l** (Figure 6). The strong electron-withdrawing effect of the furoxan ring, further enhanced by the presence of two symmetrical 1,2,4-oxadiazole scaffolds, renders both carbon atoms in the former heterocycle highly susceptible to nucleophilic attack. Consequently, two potential pathways for NO release can be envisioned, depending on whether the thiolate anion attacks the C(3) (route A) or C(4) (route B) carbon atom of the furoxan ring. At first glance, both pathways appear similar and proceed through the formation of primary intermediates **3** and **3**′, which undergo cleavage of the endocyclic N–O bond to produce anions **4** and **4**′. These intermediates subsequently decompose into nitrosoalkene **5**, accompanied by the elimination of a nitroxyl anion. The resulting species are then oxidized to generate NO, which may further oxidize to nitrite or nitrate. Owing to the presence of two identical 1,2,4-oxadiazole units, both routes converge at intermediate **5**. The key distinction between routes A and B lies in the potential cleavage of the C–N bond in intermediate **3′** in the latter pathway, which may lead to the formation of nitrosooxime **6**. This intermediate can subsequently promote the nitrosation of the thiolate anion, resulting in the generation of *S*-nitrosothiol **7**, which then decomposes to release NO while regenerating the thiol. However, based on previous reports [44,45] and the observed NO-donating properties of compounds **2a**–**k**, it is also plausible that 1,2,4-oxadiazoles undergo cleavage upon nucleophilic attack by a thiolate anion at the C(5) position of the 1,2,4-oxadiazole ring. This alternative pathway could lead to the formation of anion **8**, which would subsequently decompose into allene-like intermediate **9**, releasing a nitroxyl anion. The unstable intermediate would then undergo hydrolysis to yield hydroxysulfinylnitrile **10**.

Despite the presence of two identical 1,2,4-oxadiazole rings, the release of NO through an additional pathway is not feasible. 1,2,4-Oxadiazoles can act as NO donors through decomposition into intermediates in which nitroso groups are coupled, maintaining NO release via a 1,2,5-oxadiazolyl-type transition state. One of these nitroso groups is formed from the furoxan ring; therefore, in the presence of a second 1,2,4-oxadiazole ring, no additional nitroso group would be available beyond that formed from the same ring decomposition. This proposed mechanism accounts for the observation that compounds **2a, 2f**, and **2g** exhibited NO-donating capacities exceeding 100%, suggesting that these molecules undergo a sequential process involving NO release from both one furoxan and one 1,2,4-oxadiazole ring.

### 2.4. Anticancer Activity

For the evaluation of the anticancer properties of compounds **2a**–**l**, two MPM cell lines were selected: the murine AB1 and the human JU77 cell lines (Table 2, Appendix A). MPM was chosen as the target cancer type for two main reasons. First, the pathogenesis of MPM has been shown to be closely associated with NO, resulting in increased NO concentrations in malignant cells compared with healthy pleural tissues [46,47]. Second, structurally related mono-oxadiazole–furoxan hybrids (**III**) were previously tested under identical experimental conditions [39], enabling a direct comparison of cytotoxic effects and the establishment of structure–activity relationships.

After a 72 h incubation period, most of the tested bis-oxadiazole–furoxan hybrids exhibited micromolar cytotoxicity against both cell lines, which was either higher than or comparable to the cytotoxicity of cisplatin, a clinically approved drug used in combination with pemetrexed for MPM treatment [48]. Overall, the introduction of a second oxadiazole ring resulted in reduced cytotoxic activity compared with the previously studied 3-cyano-4-(1,2,4-oxadiazolyl)furoxans **III**, with average IC_50_ values of ~10.8 µM and ~2.6 µM for bis- and mono-oxadiazole hybrids, respectively, in the JU77 cell line [39]. Compound **2a** showed cytotoxicity in the range of 8.6–11.7 µM in both MPM cell lines, consistent with its previously reported cytotoxicity in the A549 non-small-cell lung cancer line (IC_50_ = 8.8 µM) [36]. Conversely, compound **2b** exhibited no measurable cytotoxicity, in agreement with earlier findings, likely owing to its poor solubility in cell culture media [36]. Likewise, compounds **2h** and **2i** also lacked cytotoxic effects, likely because of their aqueous solubility issues.

To elucidate structure–activity relationships, we evaluated how substituents influenced cytotoxicity in the current bis-oxadiazole series compared with the previously studied mono-oxadiazole hybrids **III** [39]. Substitution on the oxadiazole rings of compound **2a** did not consistently enhance cytotoxicity, in contrast to the mono-oxadiazole hybrids **III**, where all substitutions improved activity [39]. Nevertheless, the introduction of *o*-nitrophenyl, *o*-chlorophenyl, and furan groups consistently proved beneficial, enhancing cytotoxicity in both series against the JU77 cell line and underscoring their key role in the observed anticancer effects. The remaining derivatives displayed either comparable or reduced cytotoxicity relative to **2a.** Interestingly, several substituents that enhanced activity in hybrids **III**—such as *p*-nitrophenyl, phenyl, *p*-tolyl, and methoxyphenyl groups—led to reduced cytotoxicity in compounds **2b**, **2e**, **2h**, **2i**, and **2j**. Moreover, the pyridin-4-yl group in **2f** exhibited a dual effect, increasing cytotoxicity in AB1 cells (by 1.4-fold) while decreasing it in JU77 cells (by 1.6-fold), showing an opposite trend to that observed in the mono-oxadiazole hybrids **III**.

Although the bis-oxadiazole compounds were less cytotoxic than the mono-oxadiazole hybrids **III**, they exhibited superior selectivity toward cancer cell lines compared with non-cancerous MRC-5 lung fibroblasts. On average, the selectivity factors for bis- and mono-oxadiazole hybrids in the JU77 cell line were ~3.3 and ~1.8, respectively. Notably, compounds **2d** and **2l** exhibited remarkable 5.7–9.0-fold selectivity, suggesting a potentially broader therapeutic window that could be advantageous for clinical applications.

### 2.5. Investigation of the In Vitro Mechanism of Action

To determine whether the anticancer activity of compounds **2a–l** was linked to their NO-donating properties, we measured NO concentrations in AB1 cells treated with 25 μM of each compound for 24 h. The concentration of 25 µM was selected for mechanistic studies as it represents a pharmacologically relevant dose that is at or above the 72 h IC_50_ for the majority of the active compounds, thereby guaranteeing a measurable biological effect for cross-comparison. The incubation time was shortened to 24 h to capture early signaling events, which are more directly attributable to the primary drug mechanism rather than secondary effects of late-stage cytotoxicity. The results were compared with those of untreated control cells and expressed as percentages (Figure 7A). Compounds **2b**, **2e**, **2h**, and **2i**, which showed no cytotoxic effects in the AB1 cell line (Figure 7B), also exhibited minimal NO release. In contrast, the most cytotoxic compounds, namely **2a**, **2d**, and **2f**, produced the highest levels of intracellular NO. These findings indicate a strong correlation between cytotoxicity and intracellular NO release, suggesting that the mechanism of action of these compounds is at least partially dependent on their NO-releasing capacity.

Given the established role of NO and peroxynitrite in disrupting mitochondrial function through respiratory chain inhibition and membrane depolarization [49], we evaluated the mitochondrial toxicity of selected compounds exhibiting high (**2a**, **2f**, **2l**) and low (**2k**) NO-releasing capacity. Their effects on mitochondrial membrane potential (ΔΨm) were assessed using the JC-1 dye, a cationic probe that accumulates in mitochondria in a potential-dependent manner. At high ΔΨm, JC-1 forms red-emitting J-aggregates, whereas at low ΔΨm, it remains in its monomeric form, emitting green fluorescence. Consequently, the loss of ΔΨm is indicated by a decrease in the red-to-green fluorescence ratio, providing a quantitative measure of mitochondrial damage. AB1 cells were incubated for 24 h with 25 µM of the test compounds (**2a**, **2f**, **2l**, and **2k**), with carbonyl cyanide 3-chlorophenylhydrazone (CCCP) serving as the positive control. All tested compounds induced mitochondrial membrane depolarization, as evidenced by an increase in JC-1 monomers (Figure 8). The extent of depolarization correlated with the NO-releasing capacity of the compounds. Specifically, the low NO-releasing compound **2k** caused the least mitochondrial damage, whereas the high NO-releasing compounds **2a**, **2f**, and **2l** induced markedly higher and relatively comparable levels of depolarization.

We evaluated the ability of the selected compounds to generate reactive oxygen species (ROS) using the CM-H_2_DCFDA assay (Figure 9). In this assay, oxidation of the cell-permeable CM-H_2_DCFDA probe, mainly by H_2_O_2_, produces a fluorescent signal. AB1 cells were treated with 100 μM of compounds **2a**, **2f**, **2k**, and **2l** for 2 h. This high concentration and early time point were selected to establish ROS as a cause of the observed effects, rather than a downstream consequence. The 100 μM concentration was necessary to generate a detectable signal within this short incubation window. It was revealed that only the furan-substituted derivative **2l** caused a pronounced increase in fluorescence, indicating strong ROS generation. This finding directly implicates the furan moiety in the observed mechanism, consistent with literature reports that furan ring opening can produce toxic metabolites [50] and increase levels of hydroxyl radicals and H_2_O_2_ [51].

Subsequently, we evaluated the ability of these compounds to induce apoptosis or necrosis using the Annexin V/propidium iodide (PI) assay (Figure 10), which distinguishes early apoptotic cells (Annexin V-positive) from late apoptotic and necrotic cells (Annexin V and PI-positive). Treatment of AB1 cells with 25 µM of compounds **2a**, **2f**, **2l**, and **2k** for 24 h revealed that all compounds induced varying degrees of cell death, with efficacy directly correlating to their NO-releasing capacity. Compound **2l** caused the highest levels of both apoptosis and necrosis, followed by **2f** and **2a**. As expected, the non-NO-releasing compound **2k** did not induce significant cancer cell death.

## 3. Conclusions

In this study, we developed a convenient method for the rapid synthesis of bis(1,2,4-oxadiazolyl)furoxans using a tandem acylation/cyclization approach. The synthesized heterocyclic compounds exhibited high levels of NO release (35–75%), which were 2–3 times those of the reference furoxan-based drug candidates CAS-1609 (27%) and CHF-2363 (26%). Notably, the amount of NO released remained largely constant over the incubation period for most of the bis(1,2,4-oxadiazolyl)furoxans. The anticancer activity of the resulting bis-oxadiazole series (**2a–l**) was evaluated against murine (AB1) and human (JU77) mesothelioma cell lines. The cytotoxicity of the tested compounds was strongly influenced by the substituents on the 1,2,4-oxadiazole core, with the pyridine (**2f**), *o*-nitrophenyl (**2d**), and unsubstituted (**2a**) derivatives being the most active (IC_50_ = 3–14 µM). All compounds exhibited high selectivity toward cancer cells, with **2d** and **2l** showing 6- to 9-fold greater cytotoxicity in mesothelioma cells compared with non-cancerous human lung fibroblasts. Mechanistic studies indicated that the anticancer effects were at least partially mediated by NO donation, leading to mitochondrial damage via membrane depolarization. However, the absence of a direct correlation between NO release and apoptosis or necrosis induction suggests the involvement of additional, NO-independent cell death pathways.

## Data Availability

The original contributions presented in this study are included in the article. Further inquiries can be directed to the corresponding authors.

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
