# Peer review of "Bis-Oxadiazole Assemblies as NO-Releasing Anticancer Agents"

_pharmaceutics, 2025, doi:10.3390/pharmaceutics17111494_

Round 1
Reviewer 1 Report
Comments and Suggestions for Authors
- Matnurov et al. describe the design and synthesis of a series of bis(1,2,4-oxadiazolyl)furoxans to systematically investigate their nitric oxide (NO)-donating capacity, cytotoxicity against malignant pleural mesothelioma (MPM) cell lines, selectivity over healthy lung fibroblasts, and anticancer mechanism of action. However, only a limited number of compounds were synthesized as part of their SAR investigation, leaving significant room for a more comprehensive structure–activity relationship study.
- The figures and synthetic schemes in the manuscript should be properly formatted to fit within the text margins and maintain consistency with the overall layout.
- The cytotoxicity values presented in Table 2 should be reported with a consistent level of numerical precision to ensure clarity and facilitate accurate comparison across compounds.
- In some of the assays, relatively high concentrations (25 or 100 µM) of the best compounds were used, despite their reported low micromolar inhibition potencies. It would be helpful to clarify the rationale behind selecting these concentrations, particularly in the context of their biochemical potency and potential off-target effects at higher doses.
Author Response
Reviewer #1:
Recommendation: Publish in Pharmaceutics after minor revisions.
Comments:
We are very grateful to Reviewer #1 for the positive evaluation of our work. Please find our point-by-point replies below.
>> 1. Matnurov et al. describe the design and synthesis of a series of bis(1,2,4-oxadiazolyl)furoxans to systematically investigate their nitric oxide (NO)-donating capacity, cytotoxicity against malignant pleural mesothelioma (MPM) cell lines, selectivity over healthy lung fibroblasts, and anticancer mechanism of action. However, only a limited number of compounds were synthesized as part of their SAR investigation, leaving significant room for a more comprehensive structure–activity relationship study.
We thank the Reviewer for this insightful comment. The primary goal of the present study was to establish a comparison between the novel bis-oxadiazole furoxan scaffold and the previously explored mono-oxadiazole hybrids with similar substituents under the identical experimental conditions. We agree that expanding the chemical library is a crucial next step, and we will prioritize a more comprehensive SAR study in our future work, as suggested.
>> 2. The figures and synthetic schemes in the manuscript should be properly formatted to fit within the text margins and maintain consistency with the overall layout.
We thank the Reviewer for this comment. In fact, the size of the figures was adjusted by the production team and not by us.
>> 3. The cytotoxicity values presented in Table 2 should be reported with a consistent level of numerical precision to ensure clarity and facilitate accurate comparison across compounds.
Following Reviewer’s suggestion, the cytotoxicity values presented in Table 2 were corrected with a consistent level of numerical precision.
>> 4. In some of the assays, relatively high concentrations (25 or 100 µM) of the best compounds were used, despite their reported low micromolar inhibition potencies. It would be helpful to clarify the rationale behind selecting these concentrations, particularly in the context of their biochemical potency and potential off-target effects at higher doses.
This is an excellent point, and we thank the Reviewer for raising it. We added in the manuscript the explanation to clarify the rationale behind selecting 25 or 100 µM concentrations in different assays:
The following paragraphs have been added:
“The concentration of 25 µM was selected for mechanistic studies as it represents a pharmacologically relevant dose that is at or above the 72 h ICâ‚…â‚€ for the majority of the active compounds, thereby guaranteeing a measurable biological effect for cross-comparison. The incubation time was shortened to 24 h to capture early signaling events, which are more directly attributable to the primary drug mechanism rather than secondary effects of late-stage cytotoxicity.”
and
“This high concentration and early time point were selected to establish ROS as a cause of the observed effects, rather than a downstream consequence. The 100 μM concentration was necessary to generate a detectable signal within this short incubation window.”
Reviewer 2 Report
Comments and Suggestions for Authors
In the paper entitled "Bis-oxadiazole assemblies as NO-releasing anticancer agents", the authors synthesized 11 oxadiazole-furoxan hybrids and investigated them for their NO-releasing linked antiproliferative effects, in vitro.
While the idea of this research is good, the paper itself is blurry and misleading.
- First of all, Figure 2, compound I (one compound, but with an unknown R) is derived from 1a, 1b or 1c?...also, the O-coordinative bond is also suggesting more than one compound?;
- Scheme 1, since we have no longer 1a and 1b precursors, Figure 2 must be literature data (should be cited next to the figure);
- checked the characterization of the compounds in the Supplimentary File (SF), but it seems I could only hope that compounds 2a-k in SF are compounds 2b-l in the paper (so a is actually b);
- what are CAS-1609 and CHF-2363? for sure they are not approved drugs, but in the paper there is nothing mentioned about them.
Authors should be careful with compound numbering...otherwise paper review can be really difficult.
Comments on the Quality of English LanguageThe paper needs minor English language editing.
Author Response
Reviewer #2:
Recommendation: Publish in Pharmaceutics after minor revisions.
Comments:
We thank Reviewer #2 for their positive and encouraging assessment of our work. We have addressed their specific comments below.
>> 1. - First of all, Figure 2, compound I (one compound, but with an unknown R) is derived from 1a, 1b or 1c?...also, the O-coordinative bond is also suggesting more than one compound?;
We have revised Figure 2 to provide a clearer depiction of the molecular structure, including a specified representation of the O-coordinative bond.
>> 2. Scheme 1, since we have no longer 1a and 1b precursors, Figure 2 must be literature data (should be cited next to the figure);
We thank the reviewer for this comment. Scheme 1 has been revised to show the correct synthesis of compound 2a using the appropriate precursors, and the numbering has been updated to be consistent throughout the article.
>> 3. checked the characterization of the compounds in the Supplimentary File (SF), but it seems I could only hope that compounds 2a-k in SF are compounds 2b-l in the paper (so a is actually b);
We appreciate the Reviewer's valuable observation. The compound numbering has been corrected and is now consistent between the main article and the SF.
>> 4. what are CAS-1609 and CHF-2363? for sure they are not approved drugs, but in the paper there is nothing mentioned about them.
This is an excellent point, and we thank the Reviewer for raising it. We added in the text of manuscript more details about CAS-1609 and CHF-2363.
“For comparison, previously reported time-dependent data for compound 1b and two well-known furoxan derivatives, namely 3-carbamoyl-4-(hydroxymethyl)furoxan (CAS-1609) and 4-ethoxy-3-phenylsulfonylfuroxan (CHF-2363), were included, as these are among the most studied furoxan-based agents currently in clinical trials”
>> 5. Authors should be careful with compound numbering...otherwise paper review can be really difficult.
We thank the Reviewer for highlighting this issue and apologize for any confusion the inconsistent numbering caused. The compound numbering has now been carefully corrected and unified throughout the manuscript and SF to ensure clarity.
Additionally, following Reviewer’s suggestion to improve the English, we performed careful revision of the text to make it more readable.